# Snapshots into carbon dots formation through a combined spectroscopic approach

Francesco Rigodanza[1,5,9], Max Burian [2,6,9], Francesca Arcudi [1,7], Luka Đorđević [1,8], Heinz Amenitsch [2✉] & Maurizio Prato [1,3,4✉]

The design of novel carbon dots with ad hoc properties requires a comprehensive understanding of their formation mechanism, which is a complex task considering the number of variables involved, such as reaction time, structure of precursors or synthetic protocol employed. Herein, we systematically investigated the formation of carbon nanodots by tracking structural, chemical and photophysical features during the hydrothermal synthesis. We demonstrate that the formation of carbon nanodots consists of 4 consecutive steps: (i) aggregation of small organic molecules, (ii) formation of a dense core with an extended shell, (iii) collapse of the shell and (iv) aromatization of the core. In addition, we provide examples of routes towards tuning the core-shell design, synthesizing five novel carbon dots that all consist of an electron-dense core covered by an amine rich ligand shell.

¹ Department of Chemical and Pharmaceutical Sciences, University of Trieste, 34127 Trieste, Italy. ² Institute of Inorganic Chemistry, Graz University of Technology, Graz 8010, Austria. ³ Center for Cooperative Research in Biomaterials (CIC biomaGUNE), Basque Research and Technology Alliance (BRTA), 20014 Donostia San Sebastián, Spain. ⁴ Basque Foundation for Science, Ikerbasque, 48013 Bilbao, Spain. ⁵Present address: Department of Chemistry, University of Padova, 35151 Padova, Italy. ⁶Present address: Paul Scherrer Institute, 5232 Villigen, Switzerland. ⁷Present address: Department of Chemistry, Northwestern University, Evanston, IL 60208, USA. ⁸Present address: Simpson Querrey Institute, Northwestern University, Chicago, IL 60611, USA. ⁹These authors contributed equally: Francesco Rigodanza, Max Burian. ✉email: amenitsch@tugraz.at; prato@units.it

Carbon dots (CDs) present a unique combination of appealing properties (photoluminescence, biocompatibility, and inexpensive fabrication), such that these novel nanomaterials have already found use in optoelectronic devices, biological labeling, biomedicine, and photocatalysis[1–8]. Main efforts in the field are dedicated to (i) understanding the nature of their optical behavior and (ii) tailoring synthetic protocols so that the CDs can best comply with their applications[9–14]. However, clarifying the rather obscure steps involved in the formation of a carbon nanoparticle from small molecular precursors would be of great help in advancing the knowledge of the field and stimulate a rational synthetic approach[15–24]. Although this formation process in its entirety still remains elusive, recent work has led to hypothesize small molecular weight intermediates[25,26], helping in explaining the formation of molecular fluorophores inside the CD structure or identify heteroatom doping patterns, both usually helpful in improving the photophysical properties of the nanoparticles[26–31].

The picture is further complicated by the structural and morphological properties of the CDs, as these nanomaterials can be classified as either graphene quantum dots (GQDs), carbon quantum dots (CQDs), or carbon nanodots (CNDs). The nanoparticles (<10 nm) can have a discoidal shape (GQDs) or be quasi-spherical (CQDs and CNDs), as well as can possess (GQDs and CQDs) or not possess (CNDs) quantum confinement effects. Even though CQDs and CNDs present different properties, both are commonly prepared by bottom-up approaches from organic molecules. Contrary to top-down syntheses, the use of small organic precursors under carefully controlled synthetic conditions allows the preparation of more homogeneous and less polydisperse CDs, with smaller sizes and various functional groups on the surface. However, during CDs' bottom-up synthesis, many reactions take place simultaneously, making it difficult to monitor and define the various steps during the process.

Initially, condensation between precursors, leading to polymer formation and consequent aggregation, followed by conversion into carbogenic nanodots, through "polymerization" and "carbonization" steps was proposed[32–34]. The hypothesis of the polymer chain formation has been supported by the identification of molecules that form in the very first moments of the reaction, correlating them with the emission of the dots[35,36]. If the heating process is not efficient enough, the carbonization does not take place, yielding nanoparticles which are just aggregates of polycyclic aromatic molecules, rather than carbon nanoparticles[37,38]. On the contrary, if the heating proceeds, the evolving nanoparticles can undergo surface passivation and, if the concentration of aromatic clusters reaches the critical supersaturation point, a burst in CDs' nucleation takes place[12,39]. These nuclei then grow by aromatization of adjacent polymers, hence increasing in concentration and size at the expense of the sacrificial polymers, which finally disappear, yielding exclusively carbon nanoparticles in the reaction mixture (RM).

A fundamental contribution by Giannelis and co-workers, by a combination of TEM, TGA, FT-IR, and photoluminescence analyses, proposed a potential formation mechanism of CNDs from pyrolysis of citric acid and ethanolamine at different temperatures[40]. At first, CDs with strong photoluminescence, due to amide-containing fluorophores, are formed. As the pyrolysis proceeds, a carbogenic core appears at the expense of the molecular fluorophores, eventually yielding CNDs with photoluminescence arising only from the core. In a similar way, but investigating the formation of different CDs, Urban et al.[14] studied the evolution and the chemical nature of the fluorophores during the formation process of CQDs synthesized hydrothermally from citric acid and ethylenediamine. Importantly,

instead of varying the temperature or reagents stoichiometry, different reaction times were investigated. CQDs were formed within 30 min, maintaining the same size for the rest of the heating, while undergoing a substantial change in their internal structure with the formation of the aromatic domains that finally lead to a graphitic core. More recently, Galan and co-workers studied the formation of CNDs from glucosamine and β-alanine, with the help of $^1$H- and $^{13}$C-NMR analysis of the reaction[41]. They proposed a fast oligomerization/dimerization, followed by dehydration, aromatization (of the carbogenic core), and surface passivation. Moreover, after surface post-functionalization, the authors were able to report that the CNDs are core/shell nanoparticles: a dense core with a less-dense shell, which is formed by the molecules used for the surface post-functionalization such as glycans[42].

All these reports significantly advanced the general understanding of CDs' synthesis. Interesting is the finding that CQD and CNDs, which are different with respect to their structure and properties, appear to undergo similar transformation in the initial reaction stages. What represents the final state for CNDs, namely the amorphous core with aromatic domains, seems to be an intermediate step for CQD formation, antecedent to graphitization towards an ordered crystal lattice. Therefore, unveiling CND formation would be pivotal to understand the first stages of all CDs' growth. Specifically, a number of questions concerning CNDs and their formation remain on the table. First, it is unknown at what stage of the synthetic procedure reaction intermediates and by-products form and if their reaction pathway is independent of CNDs or vice versa. Second, little is still known about the inner structure and overall structural composition of CDs and the relative density inside the nanoparticles. And finally, the evolution of these structural aspects of CNDs during synthesis remains elusive.

We present a systematic investigation of CND formation following our reported hydrothermal synthesis from arginine (Arg) and ethylenediamine (EDA) at different reaction times[43]. These nanoparticles have been thoroughly investigated in order to understand the contribution of the single reagents to the CND formation and to prepare dots with different properties[44,45]. For example, we designed CNDs with various surface passivation in the interest of forming supramolecular structures, prepared CNDs for enhanced photocatalysis, and preserved the chirality of the reagents in the final material[43,46–48]. Herein, we monitor the evolution of the nanoparticles during the synthesis, in terms of size, shape, photoluminescence and functional groups, and identify the involved stages during the formation. Moreover, we present firm evidence to corroborate the core/shell structure of CDs and present possible synthetic approaches for tunability of core-size and layer-thickness, resulting in a series of novel CNDs.

## Results

**The synthetic protocol.** Using Arg and EDA as precursors, we produce nitrogen-doped CNDs according to our hydrothermal, microwave (MW)-assisted protocol (Methods and Supplementary Methods)[43,45]. The full synthetic procedure consists of heating at 200 W with a maximum temperature of 250 °C (Methods and Supplementary Methods) between 15 and 240 s, followed by <100 nm filtration and 48 h of dialysis (with a cut-off membrane of 1 kDa). The MW heating, at each time point, yields a crude RM that consists of three isolated species: (i) solid residue recovered on the filter (filter residue), (ii) dialysate consisting of small particles and polymers removed during dialysis (dialysate), and (iii) the final purified CNDs. Figure. 1 shows a schematic representation of synthesis, work up, and resulting compounds.

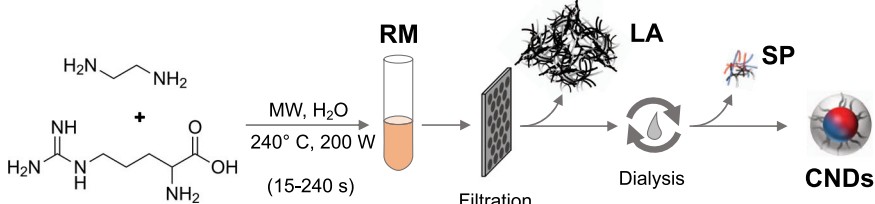

**Fig. 1 Scheme of the microwave (MW)-assisted synthetic procedure and work up.** Three constituents resulted within the reaction mixture (RM): (i) large aggregates (LA – extracted by filtration), (ii) small particles (SP – extracted by dialysis), and (iii) final carbon nanodots (CNDs).

Reaction yields (calculated based on the weight of purified CNDs after work up) over time show first a decrease from 15 to 30 s followed by an increase to a maximum of ≈25% (180–240 s) (Supplementary Fig. 1). Accordingly, the time of maximum reaction yield (after 180 s) is the reference point for comparison of experimental data in the following sections. All samples after 15 s, except CNDs, present excitation wavelength-dependent fluorescence typical for CDs, thereby confirming CND formation (Supplementary Figs. 2 and 3). Only in the first sample (after 15 s) we observe very little emission (fluorescence quantum yield (FLQY) < 0.01), indicating that this mixture contains mostly initial intermediates. In the next paragraphs we will discuss the RM and CNDs obtained from different reaction times (15–240 s), as well as the filter residue and dialysate isolated from the RM.

**Nano-structural evolution.** We track the structural phenomena during the synthesis using small- and wide-angle X-ray scattering (SAXS and WAXS, respectively), techniques commonly used to characterize both crystalline and amorphous systems at the nanoscale[49–53]. The first focus is set on SAXS and WAXS patterns of the crude RM (taken directly from the MW reactor without work up), where we expect CNDs together with reaction by-products and intermediates. The scattering data (Supplementary Fig. 4) in the first stages (15–30 s) are characterized by: (i) a clear increase in the overall scattering intensity, (ii) absence of scattering of the EDA precursor (Supplementary Fig. 5), and (iii) the onset of "aggregate scattering" (increase in low-$q$ regime ($q < 1\,nm^{-1}$), Supplementary Fig. 4). With continued heating (45–75 s), this low-angle contribution drastically increases due to the formation of a large-scale species (Supplementary Fig. 6). Dynamic light scattering (DLS) measurements confirm this observation (Supplementary Fig. 7): particles (with hydrodynamic diameter >100 nm) are present already after the first cycle (15 s), which aggregate (45–75 s) to clusters with an approximate size of 1.5 μm. WAXS patterns (Supplementary Fig. 8a) show the formation of a broad correlation peak corresponding to a mean interatomic $d$-spacing of approximately 0.37 ± 0.04 nm, as similarly found in other amorphous carbon materials, which shifts towards larger values with increasing reaction time[54,55]. We note that this evolution of the mean interatomic distance in the RM is different to the one in the isolated CNDs (Supplementary Fig. 8b), with the aggregates resulting in different sizes due to distinct concentrations between the as-obtained RM and the solution of the purified material.

Full-pattern refined model-fits of the SAXS data reveal form-factor scattering from nanoparticles <4 nm (Supplementary Methods). However, the scattering patterns of the RM should take into account all the species inside the mixture, including (i) CNDs, (ii) small by-products removed by dialysis (dialysate), and (iii) aggregates containing nanoparticles (Supplementary Fig. 6 for disentangled contributions). The SAXS pattern of the dialysate (Supplementary Fig. 9) shows that the by-products are too small to scatter in the low-angle regime. The filter residue, on the other hand, shows a SAXS pattern (Supplementary Fig. 6) reminiscent

of the aggregates mentioned above, which is in accordance with their size estimated by AFM and SEM images (Supplementary Fig. 10). However, it is unlikely that this has a big contribution to the scattering of the RM since, in order to acquire these data, we filtered 8 successive RMs and still obtained less than 1 mg of residue on the filter.

SAXS patterns were also recorded after work up (Supplementary Methods), namely samples of CNDs at 15–240 s (Fig. 2a). Here, the strong low-$q$ scattering is not anymore observed, due to the more dilute conditions, as mentioned before. Derived from refined model-fits (Fig. 2b for corresponding size distributions and Supplementary Methods for fitting details), we now detect nanoparticles at the earliest stages of 15–30 s. Then, a strong increase in the overall scattering intensity is found between CNDs at 30–45 s, indicating the onset of CND formation. Over the subsequent cycles, the size of CNDs remains nearly constant (Fig. 2b and Fig. 2c) as only a little growth is observed. Notably, the work up of the RM (filtration and dialysis) is needed to isolate the contribution of CNDs.

In addition to the structural parameters relating to size and polydispersity of CNDs, the model used for the full-pattern refined fits includes a sticky-hard-sphere interaction contribution (Supplementary Methods), yielding information on the mean interaction distance (hard-contact diameter). For CNDs, this interaction distance was found to be 1.8–2.4 nm (Fig. 2b, white trace), which is approximately 1 nm larger than the observed mean nanoparticle size. This observation results from the fact that the nanoparticle diameter (which we derive from form-factor scattering) relates to an electron-dense core of the CNDs, whereas the hard-contact (interaction) diameter includes surface ligands that are transparent to X-rays (no scattering contrast due to solvent penetration), the equivalent to a core–shell structure. Indeed, the approach of retrieving structural information from a shell that is not directly measured by X-ray scattering, but only visible by the difference of form- and structure-factor is commonly found in colloidal nanoparticle research[56,57]. Notably, this interaction distance, now relating to nanoparticles including the ligand shell, of 2.2 ± 0.2 nm is in good agreement with the average outer-size of 2.47 nm reported in our previous work (determined by statistical AFM analysis, both values relating to samples after 180 s)[43].

By subtracting the mean nanoparticle diameter from the interaction distance, the thickness of the ligand shell can be calculated (Fig. 2c, black trace). The first presence of such a shell is observed after 45 s (0.8 nm). Even though subsequent heating does not affect the inner core size (i.e., nanoparticle diameter), the shell thickness reduces to 0.4 nm between 45 and 75 s, followed by a slight increase to 0.6 nm at 240 s.

Summarizing the above findings, for the RM, SAXS and DLS data show the early (15–45 s) presence of >100 nm particle aggregates, which react (45–75 s) to form aggregates in the concentrated conditions of the RM. Most notably, SAXS data reveal in chronological order: (i) nanoparticle core and shell formation within 30–45 s, (ii) collapse of shell surfactants within

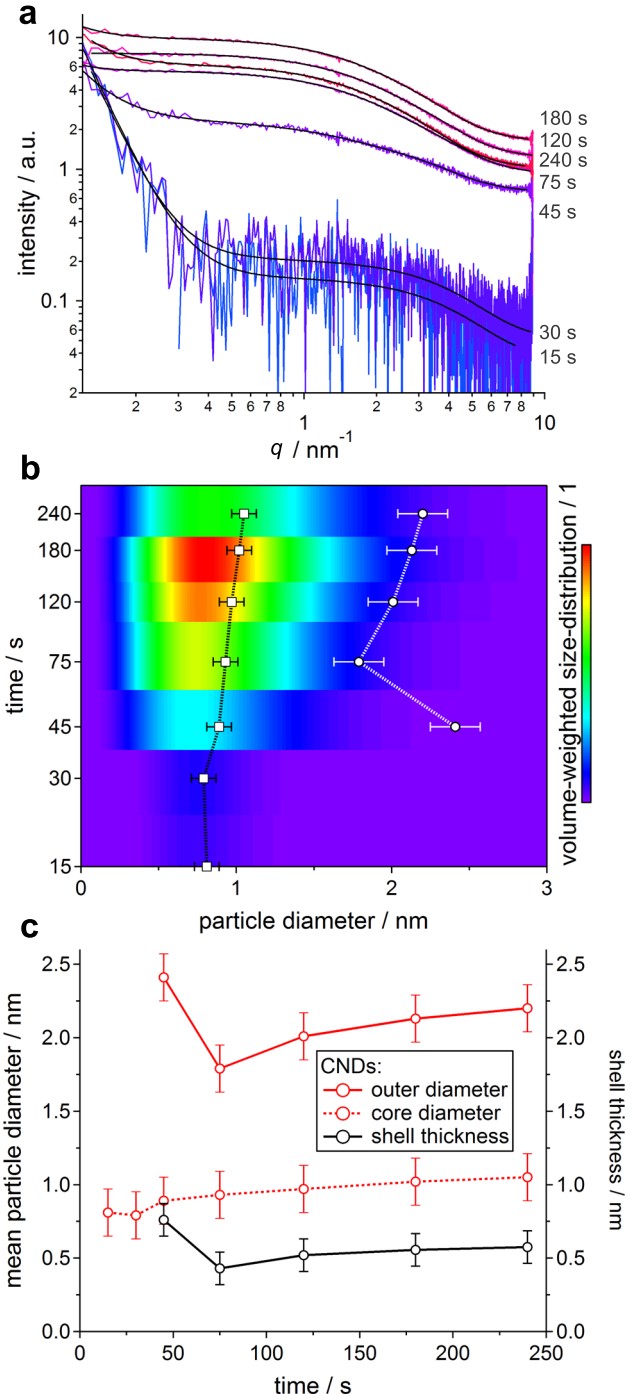

**Fig. 2 Nano-structural evolution. a** SAXS patterns and model-fits (black) of CNDs at 15–240 s. **b** Volume-weighted size distribution of CNDs, in comparison to the mean particle diameter (black trace) as well as the mean CND spacing (white trace). **c** Nanoparticle diameter as well as mean CND spacing (interaction distance–mean particle diameter) of CNDs and RM, obtained from SAXS pattern fits. Error bars denote the mean fitting uncertainty over all data sets, which is (in relative terms) <10% and <20% for fitted (core- and outer-diameter) and derived (shell thickness) parameters, respectively.

45–75 s, followed by (iii) slight particle growth and/or ripening for >75 s.

**Tracking the photophysical properties**. Generally, UV-Vis absorption spectra of CNDs present an absorption band at 290

nm, which is ascribed to the π–π* transition of conjugated C=C units (Supplementary Fig. 11)[58]. Both species, the RM and the CNDs, display a gradual increase of this absorption band with progressing reaction time, an increase not observed in filter residue. This C=C formation coincides with the first presence of nanoparticles witnessed by SAXS and occurs simultaneously with nanoparticle aggregation, previously identified by SAXS and DLS. Evolution of the C=C bond, ascribed to the aromatization of the particles, is hence likely connected to the creation of a structural anchor which, in case of CNDs, coincides with the nanoparticle core. The very same feature at 290 nm (Supplementary Fig. 11) is also detected in the spectrum of small by-products (dialysate) that we remove with dialysis[45], which could be linked to the first presence of partial CNDs.

Information on the fluorescent properties is gained from excitation-dependent emission measurements (Supplementary Figs. 2 and 3). For CNDs, the optimal excitation wavelength in the first cycles (30–45 s, as no emission is observed at 15 s) shifts from 320 to 300 nm—concomitant with (i) core formation as seen in SAXS nanoparticle and (ii) C=C bond formation as seen in UV-Vis spectra. Further heating does not appear to affect the emission properties of CNDs (75 s and onward). Notably, the early formation of the CNDs' fluorescence properties that are not affected by further aromatization is a behavior reported also by Galan et al.[41].

For the RM, emission spectra of the first reaction stages (15–75 s) undergo a similar transition, where the fluorescence from the 300 nm excitation becomes more prominent with increasing reaction time. Yet, spectra from 15–45 s contain a contribution from the 320 nm excitation: as the filter residue shows very little fluorescence (Supplementary Fig. 12), we can attribute this emission to small fluorophores with <1 kDa that are removed during dialysis and that are not resolvable by SAXS. At 75 s, emission of RM is almost identical to the purified CNDs, but after 75 s, the emissions at excitation wavelengths of 320 nm and especially 340 nm progressively become stronger. Yet different from the initial cycles, the emission of this species (after 75 s) is almost independent of the excitation wavelength (Supplementary Fig. 13) and can be ascribed to the smaller nanoparticles that we remove through dialysis.

FLQY measurements (from 300 nm excitation) support these results (Fig. 3a). For both CNDs and RM, FLQY gradually increases to 0.15 over the first 75 s and in this period the FLQY of the RM is only slightly higher than CNDs. Now in the case of CNDs, the FLQY remains constant at approximately 0.15 over the following cycles (75–240 s), consistent with unaltered emission in the same period. In the case of the RM, the increasing presence of the smallest and brightest nanoparticles (isolated FLQY of 0.45) causes a further rise of the FLQY in the following cycles, eventually reaching a value of 0.33.

In conclusion of this section, absorption/emission spectroscopy showed that the emissive sites of CNDs need little time to form (75 s). CNDs are the main emitting species inside the reaction until 75 s, when we notice new small particles with high FLQY but an almost excitation wavelength-independent emission (the origin of these particles will be discussed in the next sections).

**Chemical evolution at surface and core**. Insights into the chemical composition of the CNDs and the RM were determined by FT-IR spectroscopy and X-ray photoelectron spectroscopy (XPS). The comparison between the RM and CND FT-IR spectra (Supplementary Fig. 14) is consistent with the removal of precursors or small particles and polymers during dialysis, i.e., it shows that: (i) absorption peaks of the precursors can be recognized only in the FT-IR spectrum of the RM after the first cycle

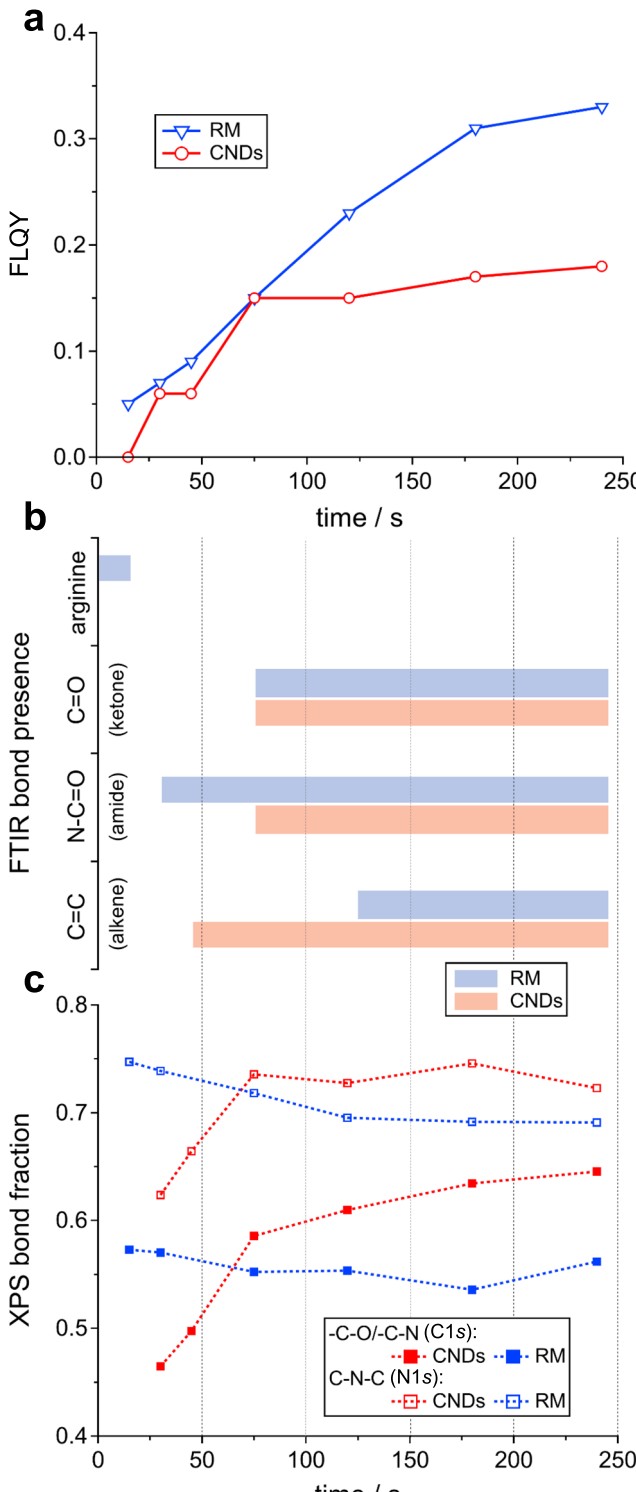

**Fig. 3 Chemical modifications over time. a** Fluorescence quantum yields (FLQY) of CNDs and RM for 15–240 s. **b** FTIR bond presence of RM and CNDs at 15–240 s. **c** XPS bond fraction (C1s and N1s) of CNDs and RM at 15–240 s.

observed in the spectra of both the RM and CNDs at longer reaction times, but in the RM and CNDs can be recognized at different points of the reaction (Fig. 3b). The chemical features mentioned above became more pronounced in the spectra of CNDs with progressing reaction time, as well as peaks at 1550 and 1456 $cm^{-1}$ that are consistent with C−N and C=N functional groups.

Quantitative analysis from XPS measurements shows a stable and consistent atomic composition of both CNDs and RMs throughout the reaction, with percentages of 62.5–67.8% for C1s, 21.7–26.8% for N1s, and 7.9–12.8% for O1s (Supplementary Fig. 15). Information on atomic-bond-composition is obtained from deconvolution of the C1s and N1s spectra (Supplementary Figs. 16 and 17). In both cases, the most dominant constituents (-C−O/-C−N for C1s and C−N−C for N1s) show strikingly different behavior for CNDs compared to the RM (Fig. 3b): while surface-near N-containing bonds in the RM do not change consistently with progressing reaction, the same bond type increases for CNDs in the early stages. Similarly, secondary peak contributions (-C−C/-C−H for C1s and -NH₂ for N1s) show an inverted behavior, indicating progressive consumption of amino groups that might be related to changes (formation, <45 s, and collapse, 45–75 s) in an N/O-rich surfactant shell observed from SAXS data.

From these findings, four main observations can be made. First, the large aggregates in the RM present different chemical features than the isolated CNDs, which is consistent with the different reaction pathway hypothesized from WAXS measurements (Supplementary Fig. 8). Indeed, a series of FT-IR absorption features (e.g., at 1490, 1280, and 1105 $cm^{-1}$) are only present in the final product of the RM but not in the CNDs. Second, the main FT-IR absorption fingerprint of CNDs is already present at 45 s, synchronous with the first observation of nanoparticles in SAXS, as well as the characteristic emission spectra. Third, the CND surface changes in bond-composition (as tracked by XPS) within the first 75 s—concomitant with the formation and collapse of the CND shell—and remains stable over the following stages. Fourth, a range of chemical transitions occur after 45 s reaction time, causing a more homogenous chemical composition, yet leaving their fluorescence properties unaffected.

**The proposed formation mechanism.** Consolidating the experimental findings above, the studied reaction results in three different species (Fig. 4): (1) large-scale aggregates >100 nm (filter residue, LA), (2) small organic particles <1 kDa (dialysate, SP), and (3) CNDs.

Regarding (1), experimental data points towards an independent reaction pathway in contrast to CNDs, as evidenced by: (a) UV-Vis and fluorescence, with the filter having a different absorption profile than CNDs and no fluorescence emission, (b) FT-IR, revealing different bond-composition in the final products of RM and CNDs, and (c) XPS, showing different bond formation kinetics in the early stages of the reaction.

Concerning (2), small organic polymers or particles form after 75 s reaction time and onwards, as seen in (i) the increasing FLQY of the crude RM as well as (ii) the evolution of an emission spectra that is independent of its excitation wavelength. In relation to their origin, it is rather unlikely that this highly emitting species stems from the completely non-fluorescent large-scale residue. A more probable scenario is that these small polymeric clusters are unformed CNDs or a product of CND degradation and/or disintegration. Indeed, obtaining 24% of final CNDs (with FLQY = 0.16 and molecular weight ≈ 1.4 kDa) and small highly fluorescent clusters (with FLQY = 0.35 and molecular weight < 1 kDa) would account for the increased FLQY in the RM (Fig. 3a) and the relatively dominant presence of the

(15 s) and not in the spectra of CNDs; (ii) the FT-IR spectra of the RM share more similarities than CNDs with the absorption peaks of the small particles and polymers removed during dialysis. The amide (C=O) stretching moves from 1695 to 1705 $cm^{-1}$ in the spectra of the RM as the reaction proceeds and the same peak at 1705 $cm^{-1}$ is observed in the spectra of the CNDs after 120 s. Peaks at 1760 $cm^{-1}$ (C=O of ketone) and 1650 $cm^{-1}$ (C=C) are

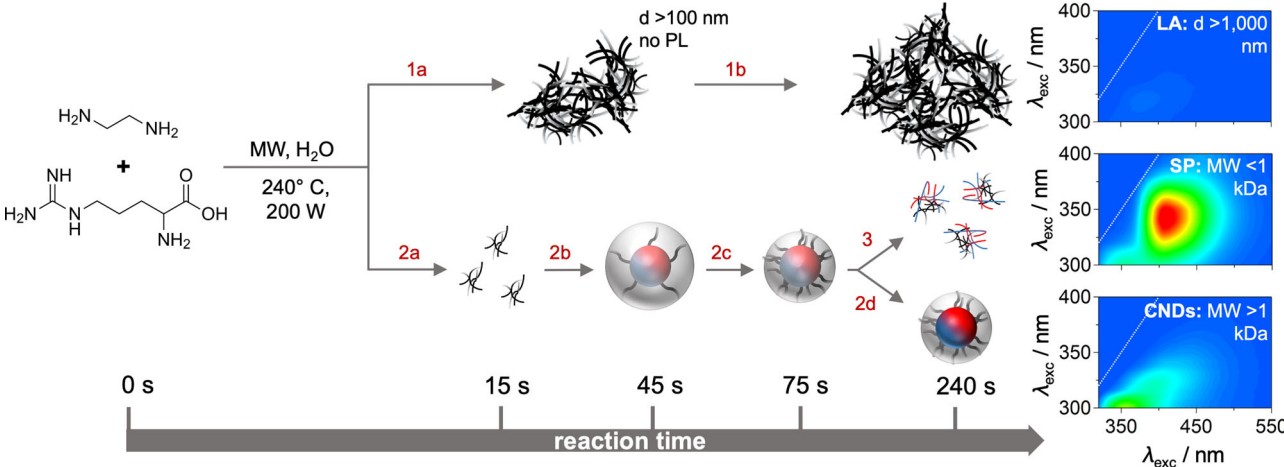

**Fig. 4 Reaction scheme.** Proposed formation of CNDs from arginine and ethylenediamine as precursors: (step 2a) aggregation of organic molecules (15–45 s), (step 2b) core and shell formation (30–45 s), (step 2c) shell collapse (45–75 s), and (step 2d) formation of aromatic groups within core (120–240 s). By-products are large aggregates >100 nm (steps 1a-1b) and small particles <1 kDa (step 3). Right insets: 2D excitation-emission spectra of filter residue (LA, top), dialysate (SP, middle), and CNDs (bottom, 0.03 mg mL$^{-1}$ in MilliQ water 25 °C), intensity is expressed in relative units.

emission at 320 and 340 nm excitation (Supplementary Fig. 13). However, and most importantly, we can exclude that these sub-nanoparticles are CNDs as we do not observe scattering of an electron-dense core in the SAXS measurements (Supplementary Figs. 10 and 18).

Regarding the formation of (3) CNDs, experimental data suggest a mechanism consisting of four consecutive steps (Fig. 4, steps 2a–2d). First (<45 s), organic molecules aggregate to form small and porous building blocks (weak SAXS intensity) that already present a low and slightly shifted excitation-dependent emission. Second (15−45 s), these organic building blocks fuse (see formation of C−N−C backbone in XPS), leading to the first presence of a carbogenic core covered by a large shell (see increase in form- and structure-factor scattering in SAXS patterns). Third (45−75 s), the shell rearranges (change in structure-factor scattering in SAXS patterns), causing a significant increase of the FLQY in CNDs (Fig. 3a). At this point, CNDs are in their final state, as excitation-dependent emission spectra and XPSdeconvolution remain unchanged from this point onward. Fourth (>75 s), CNDs slowly start to (i) disassemble into highly fluorescent organic clusters and (ii) increase aromatic domains in the CNDs' carbogenic core (see C=C formation in UV-Vis and FT-IR spectra).

**Considerations on the CNDs' core/shell-structure.** The observation of an electron-dense carbogenic core and a less-dense shell prompted us to pursue two strategies based on (i) post-synthetic modification of the shell and (ii) variation of the synthetic precursors. In the following, we select CNDs after 180 s (as previously reported) as reference (CNDs **A**; core radius = 0.54 nm, shell thickness = 0.55 nm), comparing them to structural parameters from model-fits of corresponding SAXS patterns (Supplementary Fig. 19). An overview of all involved species as well as the structural results is found in Fig. 5. The 3D fluorescence maps showing excitation-wavelength-dependent emission of all compounds can be found in Supplementary Fig. 20.

Regarding strategy (i), we intend to increase the surface shell by allowing CNDs **A** to react with Meldrum's acid (CNDs **B**). Kaiser test results show successful reaction, as the amount of primary surface-amines reduces from 1350 to 80 μmol g$^{-1}$. Indeed, the obtained core radius remains constant while the shell thickness increases by approximately 0.2 nm. This, for one, validates the hypothesis that CNDs in fact consist of an electron-dense core

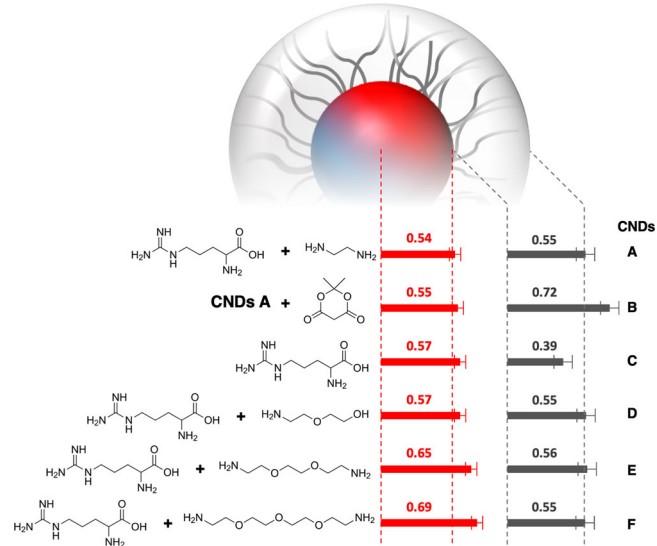

**Fig. 5 Core–shell structure of CNDs A–F as obtained from model-fits of SAXS patterns.** For better comparison, the dotted lines represent the dimensions of the reference species (CNDs **A**), which correspond to CNDs after 180 s microwave reaction with filtration and dialysis work up. Error-bars denote the mean fitting uncertainty over all data sets, which is (in relative terms) <10%.

covered by a less-dense shell, and, for the other, yields a simple example of structural tunability of CNDs.

For strategy (ii), we set to modify the core–shell ratio by repeating the same MW and work-up procedure as above (200 W, $T_{max}$ = 250 °C, 12 cycles of 15 s heating with 5 s of cooling intervals, followed by <10 nm filtration and >1 kDa dialysis) but with different (or none) diamine species. For CNDs **C**, the removal of EDA from the RM results in a negligible increase of the core radius (from 0.54 to 0.57 nm) but a strong reduction of the shell (from 0.55 to 0.39 nm). A Kaiser test value of 1050 μmol g$^{-1}$ reveals a heavy presence of primary amines in the shell that must find its origin in the arginine guanidine group. Consequently, this indicates that the oxygen-rich carboxylic moieties might play an instrumental role in the core formation. For CNDs **D**, **E**, and **F**, we hence exchange EDA with 2-(2-aminoethoxy)

ethanol, 1,8-diamino-3,6-dioxaoctane and 1,11-diamino-3,6,9-trioxaundecane, respectively (Fig. 5) to (a) increase the amount of potentially reactive ether and thereby oxo-groups and (b) increase the overall length of the amine and thereby of the potential core-building blocks (Fig. 4, stage 2a). All the new CNDs present a core/shell structure similar to CNDs **A**. For CNDs **D**, we see no mentionable change in core and shell structure. With a Kaiser test value of 490 μmol g$^{-1}$ (compared to 1350 μmol g$^{-1}$ for CNDs **A**) the primary amine concentration in the shell is rather low, likely due to the presence of alcohol groups in the surface layer. On the other hand, for CNDs **E** and **F**, we observe a gradual increase of the core region from 0.54 nm (CNDs **A**) to 0.65 and 0.69 nm (CNDs **E** and **F**, respectively). In relation to the shell region, we observe no structural changes, but only a decrease in the primary amine concentration, as Kaiser tests yield 480 μmol g$^{-1}$ for CNDs **E** and 400 μmol g$^{-1}$ for CNDs **F**. Besides the lability of the ethylene glycol chains under MW irradiation, the decrease can be reasoned by an increase in the core surface-area at constant amine concentration hence leading to decrease of the surface coverage by approximately 30% for CNDs **E** and 40% for CNDs **F**.

In summary, these structural results, first and foremost, affirm our methodology in resolving the core–shell motif in CNDs and further confirm its presence even in pristine and non-purposely functionalized CNDs. Second, we clearly see that diamines, as reaction partners, are not strictly necessary to induce such a core–shell structure, thereby emphasizing the dominant role of arginine in CND formation. Third, the results suggest that the presence of longer diamines as well as oxygen-rich moieties, particularly ethers, likely enhance the reactivity in the initial stages of CND synthesis, leading to larger molecular building blocks (Fig. 3, step 2a) and hence to increased core dimensions. In relation to the CND shell, we see no direct influence of the amine length on the effective thickness. However, post-synthetic treatment appears to be the simplest and most efficient option for structural tunability.

## Discussion

In this work, we disentangled the structural pathways of CND formation during hydrothermal MW-assisted synthesis from Arg and EDA. We demonstrated that the formation of CNDs consists of 4 consecutive structural steps: (i) aggregation of small organic molecules, (ii) formation of a dense core and with an extended shell, (iii) collapse of the shell, and (iv) aromatization of the core. Our findings reveal that the CND-unique fluorescence is achieved already after stage (iii), when CNDs present the final core–shell structure. Further heating does not influence the emissive behavior yet causes aromatization of the core. This strongly suggests that the CND shell is most responsible for their fluorescent fingerprint.

In addition, we presented five novel CNDs that all consist of an electron-dense core covered by an amine-rich ligand shell—post-synthetic CND functionalization is hence not required to form a core–shell structure. These new CNDs provide examples of possible routes towards tuning the core- and shell-size: while arginine appears to play a crucial role in templating the overall core–shell motif, addition of longer and more reactive diamines appears to affect the core dimension, whereas post-synthetic surface functionalization is likely the most efficient strategy towards shell-tunability.

Although our conclusions are derived from a specific procedure and CDs are very sensitive to reaction conditions, we found analogies with previous mechanistic studies in spite of the differences in terms of synthetic protocol and nature of

reagents[14],[40]. Thus, we believe these findings could be matched by various reported CDs once examined with the techniques we employed in this work. Therefore, future work will be directed at increasing the number of variables in the synthetic procedure to access a more rational core–shell structure which will be a step forward towards materials with specific final properties and applicability.

## Methods

**Materials synthesis**. Both CNDs and RM at 15–240 s were obtained via MW irradiation (CEM Discover-SP) of an aqueous solution of L-Arginine (Arg, Fluorochem, M03558) and ethylenediamine (EDA, Sigma-Aldrich, >99.5%, E26266). Typically, Arg (87.0 mg), EDA (33.0 μL), and Milli-Q water (100.0 μL) were heated at 200 W and 26 bar with a $T_{max}$ of 250 °C for a variable amount of time (from 15 s to 240 s). After reaching $T_{max}$ in approximately 30 s, the temperature remained constant. For RM, the powder material was obtained without any further purification by freeze-drying the mixture. For CNDs at 15–240 s, RMs were filtered through a 0.1 μm microporous membrane, washed with Milli-Q water, and dialyzed against Milli-Q water for 48 h, changing water 3 times. The powder material was obtained by freeze-drying. The synthetic procedures of CNDs B-E, LA, and SP are detailed in the Supplementary Methods.

**Material characterization**. Fourier-transform Infrared (FT-IR) spectra (KBr) were recorded on an Agilent Cary 660 FT-IR with KBr pellets spectrometer. Atomic force microscopy (AFM) images were obtained with a Nanoscope IIIa, VEECO Instruments (additional details are in Supplementary Methods). DLS measurements were performed on an Anton Paar Litesizer 500. X-ray photo-emission spectroscopy (XPS) spectra were measured on a SPECS Sage HR 100 spectrometer (additional details are in Supplementary Methods). UV-Vis spectra were recorded on a PerkinElmer Lambda 35 UV-Vis spectrophotometer. Fluorescence spectra (including 2D excitation-emission spectra) were recorded on a Varian Cary Eclipse fluorescence spectrophotometer (additional details are in Supplementary Methods). Small- and wide-angle X-ray scattering (SAXS and WAXS) measurements were performed at the Austrian SAXS beamline of the electron storage ring ELETTRA using a photon energy of 8 keV (additional details are in Supplementary Methods).

## Data availability

The data that support the findings of this study are available from the corresponding authors upon reasonable request.

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

## Acknowledgements

This work was supported by the University of Trieste, INSTM, AXA Research Fund, the Italian Ministry of Education MIUR (cofin Prot. 2017PBXPN4), the Spanish Ministry of Science, Innovation and Universities, MICIU (project PID2019-108523RB-I00), ELK-ARTEK bmG2017 (ref: Elkartek KK-2017/00008, BOPV resolution: 8 February 2018), the Maria de Maeztu Units of Excellence Program from the Spanish State Research Agency (MDM-2017-0720), and the European Research Council (ERC-AdG-2019 n. 885323). The authors acknowledge the CERIC-ERIC Consortium for the access to experimental facilities and financial support (proposal number 20182146).

## Author contributions

F.R., F.A., and L.Đ. synthesized the materials and performed optical characterization; M.B., F.R., and H.A. performed SAXS/WAXS experiments; M.B. and H.A. performed SAXS/WAXS data analyses; M.B measured and analyzed XPS and DLS experiments; L.Đ. and F.A. acquired SEM and AFM images; H.A. and M.P. designed the research; M.P. secured the funding; the manuscript was written with contributions of all authors.

## Competing interests

The authors declare no competing interests.
