## [Peer Review File · Nature Communications]

REVIEWER COMMENTS

Reviewer #1 (Remarks to the Author):

The manuscript by F. Rigodanza et al. is devoted to the revealing of the formation mechanism of the emissive carbon nanodots. The authors provided a thorough experimental analysis of the morphology, chemical composition, and optical properties of the synthesized CNDs. They also showed post-synthetic procedures which result in the controllable shift of emission with change in the CNDs structure. This manuscript can be accepted after major revision according to the comments provided below.

Major remarks

1. Please comment on the temperature of the synthesis. Since the microwave-assisted method implies heating the reaction mixture during the first 5-10 seconds of the process, how the temperature was maintained for the 15 and 30 s syntheses? Please provide this information in the part of the synthetic procedure.
2. In Figure S9 interesting results on the evolution of d-spacing in RM and CND samples during the synthesis are shown. However, it somehow contradicts the d-spacing observed for the graphene and graphite materials, like 0.21, 0.24, and 0.34 nm. The authors should provide additional information on the d-spacing values and discuss observed discrepancies. Alongside that, another x-axis should be added which shows the values of scattering degree.
3. In Figure S14b for CNDs after 75s the peak at approx. 1400 (which can be attributed to arginine) vanished with the appearance of a set of peaks in the region of 1450-1600 cm^{-1} . Please comment on this observation: to which bonds can those peaks be ascribed?
4. On page 8 authors stated: "The very same feature at 290 nm (Figure S11) is also detected in the spectrum of small byproducts (dialysate) that we remove with dialysis, which could be linked to the first presence of partial CNDs". However Figure S11 shows the absorption spectra of CNDs and RM (crude reaction mixture). Although RM contains CNDs together with dialysate and large aggregates, it is not clear how this statement is inferred from the mixture spectra.

Minor remarks

5. Line 103: the use of "herein" is quite confusing since the authors discussed previous reports.
6. Line 156: the mention of Figure S9 before Figure S8
7. Lines 185, 192, and 204: several typos in Figures numbers. Instead of "Figure 1b", "1c" "Figure 2b" and "2c" should be used. The same typo is in the Table S1 caption: "Figure 2" should be used instead of "Figure 1".
8. Line 230: after 75 "s" is missing.
9. Figure 3a: In the y axis there should be no % for FLQY.
10. In Figure S2, S12, S13, and S20 caption "fluorescence 3D matrix" should be changed to PLE-PL maps, which is more common term for such types of graphs.

11. In Figure 4 it is worth mentioning that the intensity of fluorescence in the 3D matrix is in relative units to show that the LA sample is mostly non-emissive (compared to the data provided in Figure S12).
12. Lines 311-312: here authors used "MW" for molecular weight instead of designated above "microwave". Please, make it clear.
13. Check the writing of "ad-hoc", "by-product", "work-up", etc.

Reviewer #2 (Remarks to the Author):

The manuscript by Rigodanza et al. describes a very detailed and interesting study of carbon nanodot formation from precursor solutions of arginine and ethylenediamine via hydrothermal microwave-assisted methods. The work smartly combines X-ray scattering methods with fluorescence and spectroscopic probes of chemical composition to understand the evolution of both structure and chemistry during CND synthesis. Their results lead to a proposed mechanism of CND formation that involves formation of small organic aggregates and oligomers followed by core formation, shell collapse and subsequent aromatization.

The manuscript is thorough and nicely combines information from multiple methods to achieve a complete mechanistic picture. A series of experiments that explores the synthesis of modified CNDs by changing the composition of the precursor solution is also used to put the mechanistic hypothesis to the test. I believe publication in this journal is justified, given the intense interest in CNDs as nanomaterials for a range of applications, particularly bioimaging, and the importance of controlling their synthesis and properties in a rational manner through mechanistic understanding. I am therefore happy to recommend publication with some minor comments and changes as follows:

- 1) The abstract should be improved, as it currently provides little information and no quantitative results, only a description of the methods used. I would suggest redrafting.
- 2) Page 8 line 204; the reference to Figure 1c should be Figure 2c instead. Page 10, line 258: a reference to "next conclusive chapters" should be amended, e.g. "next sections".
- 3) Figure 3b: it is not clear what criterion was used to state presence/absence of peaks in the FTIR. When examining Figure S14 for instance, one could argue that the C=O peak is not resolved before 75s in the CND signal, whereas in Figure 3b it is reported as present at 45s. Similarly, for the CND in Fig.S14 there is an arginine peak present until 75 s but it is shown as absent in Figure 3b. Greater clarity in the experimental section on how presence/absence is established, and a consistency check between Fig. 3b and S14 is recommended.
- 4) The authors explore the applicability of their mechanistic hypothesis by using a range of precursor solutions with different compositions. However it is less clear how general is the proposed mechanism

in relation to the synthesis conditions; could the authors comment or expand on whether e.g. batch size/MW power should affect the observed evolution of organic aggregates, core and shell?

Reviewer #3 (Remarks to the Author):

The paper reports a systematic investigation of carbon nanodots (CNDs) formation through a combined spectroscopic approach. The synthesis, characterization, and reaction mechanism of CNDs have received much attention. The authors collected a great deal of the data and build discussions for the structure, formation mechanism, and characterization of CNDs based on the spectroscopic results. Thus, the paper is useful and important for the community of the CNDs. The reviewer can recommend publish if the authors consider the following minor points.

(1) The yields of the small particles (SP) and the large aggregates (LA) were not shown.

(2) The quantum yields (QY) of the reaction mixture (RM) at 75 s was almost identical to the QY of CNDs, which indicate the emission from RM was due to the emission from CNDs. Although QY of CNDs was not increased after 75 s, the QY of RM was increased after 75 s. Therefore, the analysis of SP is necessary for the further understanding, such as the absorption and emission spectra. The excitation wavelength for QY measurements was not shown.

(3) The absorption and emission intensities are important for the quantitative understanding. How were they defined?

(4) The particle size reached 1.5 μm after 240 s (p. 5 L155, Figure S7(b)).

(5) Figure 1c (p.5, L185, p.6, L204) might be Figure 2c.

We thank the referees for their time, comments and the opportunity given us to clarify and correct some aspects of our work. Consequently, we believe that the status of the present manuscript has been improved over the reviewing process.

Reviewer 1.

The manuscript by F. Rigodanza et al. is devoted to the revealing of the formation mechanism of the emissive carbon nanodots. The authors provided a thorough experimental analysis of the morphology, chemical composition, and optical properties of the synthesized CNDs. They also showed post-synthetic procedures which result in the controllable shift of emission with change in the CNDs structure. This manuscript can be accepted after major revision according to the comments provided below.

Answer: We thank the Reviewer for considering our manuscript suitable for publication in Nature Communications. We have carefully addressed the helpful comments and believe that the manuscript has improved.

1) Please comment on the temperature of the synthesis. Since the microwave-assisted method implies heating the reaction mixture during the first 5-10 seconds of the process, how the temperature was maintained for the 15 and 30 s syntheses? Please provide this information in the part of the synthetic procedure.

Answer: We thank the Reviewer for his/her careful comment. Accordingly, we described in more detail the microwave-assisted method utilized. As the Reviewer correctly stated, the reaction requires an initial heating time before reaching the temperature of 250 °C (which is the T_{max}). In our case, the temperature reached the maximum temperature (and remained constant) after approximately 30 seconds (2nd cycle). As mentioned in the revised manuscript, the starting materials (arginine) can still be observed in the reaction mixture after 15 seconds, and this is most probably due to the vessel not yet reaching the maximum temperature.

We have added these details in the section “Materials and Synthesis” of the Supplementary Information and we have referred the readers to our recent Nature Protocol paper (reference 46 in the originally submitted manuscript, *Nat. Protoc.* **14**, 2931–2953 (2019)) that includes a graph with the evolution of temperature, pressure, and power, during the CNDs reaction time.¹

2) In Figure S9 interesting results on the evolution of d-spacing in RM and CND samples during the synthesis are shown. However, it somehow contradicts the d-spacing observed for the graphene and graphite materials, like 0.21, 0.24, and 0.34 nm. The authors should provide additional information on the d-spacing values and discuss observed discrepancies. Alongside that, another x-axis should be added which shows the values of scattering degree.

Answer: Indeed, the reviewer raises a good point on the accuracy and meaning of the obtained inter-atomic d-spacings observed in the WAXS data. The mentioned discrepancy mainly stems from the fact that the coherent scattering volume of CNDs is very small. This coherent scattering volume relates inversely to the width of scattering peaks in XRD according to the Debye-Scherrer-Equation: for a CND core of approximately 1 nm diameter, one would expect a peak width of $2\pi/d \approx 6 \text{ nm}^{-1}$ in reciprocal space (see e.g. the WAXS pattern of the pure CNDs shown in Supplementary Figure 8b). Due to this broad peak shape, we cannot determine the interatomic d-spacing with a precision better than approximately 10% and we can only use the experimental data as an indication of the mean interatomic distance. Indeed, XRD spectra of e.g. amorphous carbon materials where a similar coordination arrangement is to be expected (e.g. see Fig. 4 in *Energy Environ. Sci.*, 2014,7, 1110-1116 or see Fig. 1 in *Sci. Bull.*, 2018, 63, 1125-1129)^{2,3} present the same broad peak shape, corresponding to a d-spacing of approx. 0.35 nm – in very good agreement with our observation. In short, our WAXS measurements suggest and reaffirm the amorphous nature of the CND core. We have now commented this in the main text and, for better comparison with literature data, show a second 2-theta axis in the Supplementary Figures.

The manuscript was updated as follows:

“WAXS patterns (Supplementary Fig. 8a) show the formation of a broad correlation peak corresponding to a mean interatomic d-spacing of approx. $0.37 \pm 0.04 \text{ nm}$, as similarly found in other amorphous carbon materials, which shifts towards larger values with increasing reaction time. We note that this evolution of the mean interatomic distance in the RM is different to the one in the isolated CNDs (Supplementary Fig. 8b), with the aggregates resulting in different sizes due to distinct concentrations between the as obtained reaction mixture and the solution of the purified material.”

3) In Figure S14b for CNDs after 75s the peak at approx. 1400 cm^{-1} (which can be attributed to arginine) vanished with the appearance of a set of peaks in the region of $1450\text{-}1600 \text{ cm}^{-1}$. Please comment on this observation: to which bonds can those peaks be ascribed?

Answer: We thank the Reviewer for his/her comment about the IR spectra in Supplementary Fig. 14. As the reviewer correctly pointed out, the peaks typical of the arginine disappear during the heating process. As suggested by the reviewer, we have now attributed the peaks at 1550 and 1456 cm^{-1} to C-N and C=N functional groups. We have amended the main text accordingly and we have also included the comparison with the spectrum of the dialysate in the discussion (and the data in Supplementary Fig.14).

The manuscript was updated as follows:

“The comparison between the RM and CNDs FTIR spectra (Supplementary Fig. 14) is consistent with the removal of precursors or small particles and polymers during dialysis, *i.e.* it shows that: (i) absorption peaks of the precursors can be recognized only in the FTIR spectrum of the RM after the first cycle (15 s) and not in the spectra of CNDs; (ii) the FTIR spectra of the RM share more

similarities than CNDs with the absorption peaks of the small particles and polymers removed during dialysis. The amide (C=O) stretching moves from 1695 cm^{-1} to 1705 cm^{-1} in the spectra of the RM as the reaction proceeds and the same peak at 1705 cm^{-1} is observed in the spectra of the CNDs after 120 s. Peaks at 1760 cm^{-1} (C=O of ketone) and 1650 cm^{-1} (C=C) are observed in the spectra of both the RM and CNDs at longer reaction times, but in the RM and CNDs can be recognized at different points of the reaction (Figure 3b). The chemical features mentioned above became more pronounced in the spectra of CNDs with progressing reaction time, as well as peaks at 1550 and 1456 cm^{-1} that are consistent with C-N and C=N functional groups.”

4) On page 8 authors stated: "The very same feature at 290 nm (Figure S11) is also detected in the spectrum of small byproducts (dialysate) that we remove with dialysis, which could be linked to the first presence of partial CNDs". However, Figure S11 shows the absorption spectra of CNDs and RM (crude reaction mixture). Although RM contains CNDs together with dialysate and large aggregates, it is not clear how this statement is inferred from the mixture spectra.

Answer: We thank the referee for his/her suggestion. We apologize for the confusion and for not including the spectra in the original submission. We amended the Supplementary Information to include the absorption spectrum of small byproducts (SP) and large aggregates (LA). These spectra were previously published in a separate work focused on the preparation and characterization of carbon nanodots (reference 46 in the originally submitted manuscript, *Nat. Protoc.* **14**, 2931–2953 (2019)) and herein replotted in Supplementary Fig. 11 for clarity.¹

We amended the main text as follow:

“The very same feature at 290 nm (Supplementary Fig. 11) is also detected in the spectrum of small byproducts (dialysate) that we remove with dialysis,⁴⁶ which could be linked to the first presence of partial CNDs”.

Supplementary Fig. 11. a) Normalized (200 nm) UV-Vis spectra of RM (15-240 s - H₂O, 298 K). b) Normalized UV-Vis spectra of CNDs (30-240 s - H₂O, 298 K). c) Normalized UV-Vis spectra of SP and LA (H₂O, 298 K) replotted from ref 1.¹ In all measurement series, the clear increase of the 290 nm absorption band corresponding to C=C formation is observed, as highlighted in the corresponding insets.

Reviewer 2.

The manuscript by Rigodanza et al. describes a very detailed and interesting study of carbon nanodot formation from precursor solutions of arginine and ethylenediamine via hydrothermal microwave-assisted methods. The work smartly combines X-ray scattering methods with fluorescence and spectroscopic probes of chemical composition to understand the evolution of both structure and chemistry during CND synthesis. Their results lead to a proposed mechanism of CND formation that involves formation of small organic aggregates and oligomers followed by core formation, shell collapse and subsequent aromatization.

The manuscript is thorough and nicely combines information from multiple methods to achieve a complete mechanistic picture. A series of experiments that explores the synthesis of modified CNDs by changing the composition of the precursor solution is also used to put the mechanistic hypothesis to the test. I believe publication in this journal is justified, given the intense interest in CNDs as nanomaterials for a range of applications, particularly bioimaging, and the importance of controlling their synthesis and properties in a rational manner through mechanistic understanding. I am therefore happy to recommend publication with some minor comments and changes as follows:

Answer: We thank the Reviewer for his/her kind words. We are delighted that the main points of this work were recognized, and we welcome the suggestions from the referee.

1) The abstract should be improved, as it currently provides little information and no quantitative results, only a description of the methods used. I would suggest redrafting.

Answer: We thank the referee for his/her suggestion. We have revised the abstract in the following way:

The design of novel carbon dots with ad hoc properties requires a comprehensive understanding of their formation mechanism, which is a complex task considering the number of variables involved, such as reaction time, structure of precursors or synthetic protocol employed. Herein, we systematically investigated the formation of carbon nanodots by tracking structural, chemical and photophysical features during the hydrothermal synthesis. We demonstrate that the formation of carbon nanodots consists of 4 consecutive steps: *i*) aggregation of small organic molecules, *ii*) formation of a dense core with an extended shell, *iii*) collapse of the shell and *iv*) aromatization of the core. In addition, we provide examples of routes towards tuning the core-shell design, synthesizing five novel carbon dots that all consist of an electron dense core covered by an amine rich ligand shell.

2) Page 8 line 204; the reference to Figure 1c should be Figure 2c instead. Page 10, line 258: a reference to "next conclusive chapters" should be amended, e.g. "next sections".

Answer: We thank the referee for his/her suggestion and the main text was amended accordingly.

3) Figure 3b: it is not clear what criterion was used to state presence/absence of peaks in the FTIR. When examining Figure S14 for instance, one could argue that the C=O peak is not resolved before 75s in the CND signal, whereas in Figure 3b it is reported as present at 45s. Similarly, for the CND in Fig.S14 there is an arginine peak present until 75 s but it is shown as absent in Figure 3b. Greater clarity in the experimental section on how presence/absence is established, and a consistency check between Fig. 3b and S14 is recommended.

Answer: We thank the referee for his/her observation. We agree with him/her and we have now corrected the Figures reporting the FTIR data. Following the comments from the referees, we have also included the data of the dialysate including the FTIR (in the new Supplementary Fig. 14). To improve the clarity related to the attributions of the peaks, we have now better highlighted the relevant absorptions in the new Supplementary Fig. 14.

The manuscript was updated as follows:

“The comparison between the RM and CNDs FTIR spectra (Supplementary Fig. 14) is consistent with the removal of precursors or small particles and polymers during dialysis, *i.e.* it shows that: (i) absorption peaks of the precursors can be recognized only in the FTIR spectrum of the RM after the first cycle (15 s) and not in the spectra of CNDs; (ii) the FTIR spectra of the RM share more similarities than CNDs with the absorption peaks of the small particles and polymers removed during dialysis. The amide (C=O) stretching moves from 1695 cm^{-1} to 1705 cm^{-1} in the spectra of the RM as the reaction proceeds and the same peak at 1705 cm^{-1} is observed in the spectra of the CNDs after 120 s. Peaks at 1760 cm^{-1} (C=O of ketone/anhydride) and 1650 cm^{-1} (C=C) are observed in the spectra of both the RM and CNDs at longer reaction times, but in the RM and CNDs can be recognized at different points of the reaction (Figure 3b). The chemical features mentioned above became more pronounced in the spectra of CNDs with progressing reaction time, as well as peaks at 1550 and 1456 cm^{-1} that are consistent with C-N and C=N functional groups.”

4) The authors explore the applicability of their mechanistic hypothesis by using a range of precursor solutions with different compositions. However, it is less clear how general is the proposed mechanism in relation to the synthesis conditions; could the authors comment or expand on whether *e.g.* batch size/MW power should affect the observed evolution of organic aggregates, core and shell?

Answer: We thank the Reviewer for his/her interest in the topic and the suggestions about wider method applicability. Carbon dots are very sensitive to synthetic conditions, so parameters like size, power and, temperature have a great impact on the final properties of the material. However, regarding the mechanism, we observed many analogies with what was reported by other authors, despite the differences in the synthetic protocol and nature of reagents.⁴⁻⁶ Indeed, the key steps of an initial polymerization/condensation of the precursors and a later carbonization/aromatization are recurrent in literature. Moreover, even if CQD and CNDs are different with respect to their structure

and properties, they appear to undergo similar transformation in the initial reaction stages. What represents the final state for CNDs, namely the amorphous core with aromatic domains, seems to be an intermediate step for CQD formation, antecedent to graphitization towards an ordered crystal lattice.

However, being the first work on the evolution of a core-shell structure during the CNDs formation, it is difficult to predict if and how this structure is repeated when changing drastically the synthesis conditions. Our current work is focused on expanding these parameters to increase the variability of the CNDs properties. Thanks to Reviewer suggestions a starting point for future research, we amended the “conclusions” chapter to include these observations.

The manuscript was updated as follows:

“Although our conclusions are derived from a specific procedure and carbon dots are very sensitive to reaction conditions, we found analogies with previous mechanistic studies in spite of the differences in terms of synthetic protocol and nature of reagents.^{4,5} Thus, we believe these findings could be matched by various reported carbon dots once examined with the techniques we employed in this work. Therefore, future work will be directed at increasing the number of variables in the synthetic procedure to access a more rational core-shell structure which will be a step-forward towards materials with specific final properties and applicability.”

Reviewer 3.

The paper reports a systematic investigation of carbon nanodots (CNDs) formation through a combined spectroscopic approach. The synthesis, characterization, and reaction mechanism of CNDs have received much attention. The authors collected a great deal of the data and build discussions for the structure, formation mechanism, and characterization of CNDs based on the spectroscopic results. Thus, the paper is useful and important for the community of the CNDs. The reviewer can recommend publish if the authors consider the following minor points.

Answer: We thank the Reviewer for his/her kind words. We were pleased to address (and include in the revised manuscript) the suggestions from the referee.

1) The yields of the small particles (SP) and the large aggregates (LA) were not shown

Answer: We thank the Reviewer for his/her observations. We updated the “Materials and Synthesis” Section in the Supplementary Information with the details of SP and LA purification and we have referred the readers to our recent Nature Protocol (reference 46 in the originally submitted manuscript, *Nat. Protoc.* **14**, 2931–2953 (2019)), which included part of this information when we detailed the purification and characterization procedures of carbon dots.¹

Regarding the yield of these fractions, we have attempted to quantify the yield of the SP, or dialysates, after each reaction, but this posed some issues. First of all, to obtain enough material for characterization of LA, we performed 8 consecutive synthesis and passed them through the same filter. Even in this case, we obtained just under 1 mg of LA (0.9 mg, <1% yield), that was however enough to perform the photophysical, microscopic and solution X-ray scattering characterizations (reported as Supplementary Figures 6, 9, 11 and 12). In the case of SP (and dialysates in general), we noticed a progressive change of color of the material (from light yellow to dark brown) during their handling (dialysate recovery, rotary evaporation of the aqueous solutions and freeze-drying) which would lead to incorrect characterizations (or conclusions) regarding the progress of the reaction. For this reason, we have collected only the dialysates from the first water changes and used those for characterization purposes. Therefore, these fractions were isolated to have a qualitative comparison with the CNDs purified over time.

The details of SP and LA procedure in the Supplementary Information have been modified as follows:

LA were obtained via microwave irradiation of an aqueous solution of L-Arginine and Ethylenediamine (EDA) (0.5 mmol). Typically, L-Arginine (87.0 mg), EDA (33.0 μ L) and Milli-Q water (100.0 μ L) were heated at 200 W and 26 bar with a T_{\max} of 250 $^{\circ}$ C for 16 cycles of 15 s heating and 5 s cooling. To obtain enough material for characterization of LA, we performed 8 consecutive syntheses and passed them on the same filter (0.1 μ m microporous membrane), which was sonicated with water (5 mL) and finally freeze-dried. Even in this case, we obtained 0.9 mg of LA (<1% yield), which was however enough to perform the photophysical, microscopies and solution X-ray scattering characterizations (reported in the Supplementary Figs. 6, 9, 11 and 12).

SP were obtained via microwave irradiation of an aqueous solution of L-Arginine and Ethylenediamine (EDA) (0.5 mmol). Typically, L-Arginine (87.0 mg), EDA (33.0 μ L) and Milli-Q water (100.0 μ L) were heated at 200 W and 26 bar with a T_{max} of 250 $^{\circ}$ C for 16 cycles of 15 s heating and 5 s cooling. The reaction mixtures were filtered (0.1 μ m microporous membrane), washed with Milli-Q water, and dialyzed against Milli-Q water. Regarding the yield of these fractions, we have attempted to quantify them, but this procedure posed some issues. We noticed a progressive change of color of the material (from light yellow to dark brown) during their handling (dialysate recovery, rotary evaporation of the aqueous solutions and freeze-drying) which would lead to incorrect characterizations (or conclusions) regarding the progress of the reaction. For this reason, we have collected only the dialysates from the first two water changes (first 8 hours), concentrated them, and used those for characterization purposes. Therefore, for the photophysical studies these fractions were used to have a qualitative comparison with the CNDs purified over time.

2) *The quantum yields (QY) of the reaction mixture (RM) at 75 s was almost identical to the QY of CNDs, which indicate the emission from RM was due to the emission from CNDs. Although QY of CNDs was not increased after 75 s, the QY of RM was increased after 75 s. Therefore, the analysis of SP is necessary for the further understanding, such as the absorption and emission spectra. The excitation wavelength for QY measurements was not shown.*

Answer: We thank the Reviewer for his/her comment. He/she raises a good point on the attribution of the emission after 75 s, which was a key point during our investigation. Previously, we have characterized the small nanoparticles (SP) at the end of the microwave irradiation (referred to as 'dialysate' in that Nat. Protocols; reference 46 in the originally submitted manuscript, *Nat. Protoc.* **14**, 2931–2953 (2019)) and they were characterized in terms of absorption, emission and FLQY. We updated the Supplementary Information including these pieces of information that were replotted in Supplementary Fig. 11-13 for clarity.¹ Moreover, we added the excitation wavelength used for the FLQY measurements (320 nm) in the SI section "Methods".

Supplementary Fig. 11. a) Normalized (200 nm) UV-Vis spectra of RM (15-240 s – H₂O, 298 K). b) Normalized UV-Vis spectra of CNDs (30–240 s – H₂O, 298 K). c) Normalized UV-Vis spectra of SP and LA (H₂O, 298 K) replotted from ref 1.¹ In all measurement series, the clear increase of the 290 nm absorption band corresponding to C=C formation is observed, as highlighted in the corresponding insets.

Supplementary Fig. 12. Fluorescence spectra of the LA recovered from the filter during work up of the RM after 180 s reaction (H₂O, 298 K).

Supplementary Fig. 13. Fluorescence spectra of the SP recovered from the dialysate during work up of the RM after 180 s reaction (H₂O, 298 K).

3) *The absorption and emission intensities are important for the quantitative understanding. How were they defined?*

Answer: We thank the Reviewer for the possibility of making clearer the treatment of experimental data. We reported the emission intensities in the spectra as arbitrary units (a.u.) since we calculated the fluorescence quantum yields for each RM and CNDs sample. This information has been added in the Supplementary Information (section “Methods”): “Emission intensities are reported as arbitrary units (a.u.) since the FLQY was calculated for each sample (reported in Fig. 3a).” Regarding the absorption intensities and the quantification of RM and SP, we run into the same issues outlined before for their yields (see reply to point 1). In short, we noticed a progressive change of color of the material (from light yellow to dark brown) during their separation and isolation. Therefore, the collected dialysates were concentrated and had to be used for a qualitative comparison (and are reported as normalized intensities).

References

1. Đorđević, L., Arcudi, F. & Prato, M. Preparation, functionalization and characterization of engineered carbon nanodots. *Nat. Protoc.* **14**, 2931–2953 (2019).
2. Rajan, A. S., Sampath, S. & Shukla, A. K. An in situ carbon-grafted alkaline iron electrode for iron-based accumulators. *Energy Environ. Sci.* **7**, 1110–1116 (2014).
3. Zhao, C. *et al.* High-temperature treatment induced carbon anode with ultrahigh Na storage capacity at low-voltage plateau. *Sci. Bull.* **63**, 1125–1129 (2018).
4. Krysmann, M. J., Kellarakis, A., Dallas, P. & Giannelis, E. P. Formation mechanism of carbogenic nanoparticles with dual photoluminescence emission. *J. Am. Chem. Soc.* **134**, 747–750 (2012).
5. Ehrat, F. *et al.* Tracking the Source of Carbon Dot Photoluminescence: Aromatic Domains versus Molecular Fluorophores. *Nano Lett.* **17**, 7710–7716 (2017).
6. Swift, T. A. *et al.* Surface functionalisation significantly changes the physical and electronic properties of carbon nano-dots. *Nanoscale* **10**, 13908–13912 (2018).

REVIEWERS' COMMENTS

Reviewer #1 (Remarks to the Author):

The authors addressed all the reviewer remarks and suggestions in an adequate way. I can now recommend this manuscript to be published.

Reviewer #3 (Remarks to the Author):

The authors carefully consider these reviewers comment and the manuscript has been revised well. I think this manuscript will be accepted in its current form.